# SARS-CoV-2 Infection in Pregnancy: Placental Histomorphological Patterns, Disease Severity and Perinatal Outcomes

**DOI:** 10.3390/ijerph19159517

**Published:** 2022-08-03

**Authors:** Yin Ping Wong, Geok Chin Tan, Siti Zarqah Omar, Muaatamarulain Mustangin, Yogesh Singh, Madhuri S. Salker, Nor Haslinda Abd Aziz, Mohamad Nasir Shafiee

**Affiliations:** 1Department of Pathology, Faculty of Medicine, Universiti Kebangsaan Malaysia, Jalan Yaacob Latif, Bandar Tun Razak, Kuala Lumpur 56000, Malaysia; amar@ppukm.ukm.edu.my; 2Department of Pathology, Hospital Sultanah Nur Zahirah, Kuala Terengganu 20400, Terengganu, Malaysia; zarqahomar@gmail.com; 3Institute of Medical Genetics and Applied Genomics, University of Tubingen, Calwerstrasse 7, 72076 Tubingen, Germany; yogesh.singh@med.uni-tuebingen.de; 4Research Institute for Women’s Health, University of Tubingen, Calwerstrasse 7, 72076 Tubingen, Germany; madhuri.salker@med.uni-tuebingen.de; 5Department of Obstetrics and Gynaecology, Faculty of Medicine, Universiti Kebangsaan Malaysia, Jalan Yaacob Latif, Bandar Tun Razak, Kuala Lumpur 56000, Malaysia; norhaslinda.abdaziz@ppukm.ukm.edu.my (N.H.A.A.); nasirshafiee@hotmail.com (M.N.S.)

**Keywords:** COVID-19, histomorphology, maternal death, neonatal outcomes, placenta, pregnancy, SARS-CoV-2

## Abstract

The association between maternal COVID-19 infection, placental histomorphology and perinatal outcomes is uncertain. The published studies on how placental structure is affected after SARS-CoV-2 virus in COVID-19-infected pregnant women are lacking. We investigated the effects of maternal SARS-CoV-2 infection on placental histomorphology and pregnancy outcomes. A retrospective cohort study on 47 pregnant women with confirmed SARS-CoV-2 infection, matched with non-infected controls, was conducted. Relevant clinicopathological data and primary birth outcomes were recorded. Histomorphology and SARS-CoV-2 immunohistochemistry analyses of placental tissues were performed. Only 1 of 47 cases showed SARS-CoV-2 immunoreactivity in the syncytiotrophoblasts. Histologically, decidual vasculopathy (n = 22/47, *p* = 0.004), maternal vascular thrombosis (n = 9/47, *p* = 0.015) and chronic histiocytic intervillositis (n = 10/47, *p* = 0.027) were significantly higher in the COVID-19-infected placentas when compared to the control group. Maternal vascular thrombosis was a significant feature in the active COVID-19 group. A significant lower gestational age (*p* < 0.001)) at delivery and a higher caesarean section rate (*p* = 0.007) were observed in the active SARS-CoV-2-infected cases, resulting in a significant lower fetal-placental weight ratio (*p* = 0.022) and poorer Apgar score (*p* < 0.001). Notably, active (*p* = 0.027), symptomatic (*p* = 0.039), severe-critical (*p* = 0.002) maternal COVID-19 infection and placental inflammation (*p* = 0.011) were associated with an increased risk of preterm delivery. Altered placental villous maturation and severe-critical maternal COVID-19 infection were associated with an elevated risk of poor Apgar scores at birth (*p* = 0.018) and maternal mortality (*p* = 0.023), respectively.

## 1. Introduction

Coronavirus disease 19 (COVID-19), a novel zoonotic viral disease, is caused by the notorious severe acute respiratory syndrome coronavirus 2 (SARS-CoV-2). This potentially fatal lower respiratory tract infection has developed into a pandemic wave ravaging many countries worldwide since its first encounter in Wuhan province, China, in December 2019. As of 28 July 2022, over 570 million people across 225 countries were infected with more than 6.3 million confirmed deaths, amounting to a 1.1% death rate [1].

Increasing evidence suggests that pregnant women do not seem to be at a heightened risk of infection of SARS-CoV-2 nor display a more severe disease manifestation compared to non-pregnant women [2]. The majority of pregnant women infected with SARS-CoV-2 are asymptomatic or only experience mild symptoms [2]. The impact of COVID-19 on symptomatic gravid individuals, however, has been greater than on non-pregnant women of similar age, with more frequent intensive care unit (ICU) admission, mechanical ventilatory, and circulatory support dependency [3,4,5]. A cohort study in England revealed that pregnant women who contracted the virus were far more likely to have a stillbirth and other adverse outcomes such as higher rates of pre-eclampsia, birth by emergency caesarean delivery, and preterm birth [6].

Certain viruses such as herpes simplex virus, rubella, cytomegalovirus, and Zika are notoriously associated with stillbirth when contracted during pregnancy, mostly through intrauterine fetal infection that induces birth defects [7]. While most neonates born to COVID-19-positive mothers tested negative for SARS-CoV-2, an estimated 3% and 13% of newborns tested positive for SARS-CoV-2 at birth and within 48 h of birth, respectively [8], and most reported no or early-onset symptoms. Whether this is due to the transplacental transmission or through environmental exposure following delivery is unclear [9]. So far, no teratogenic effects have been documented in the SARS-CoV-2-positive neonates [9]. The mechanisms behind the increased stillbirths in SARS-CoV-2-infected pregnant women nevertheless remain a mystery.

Despite being a respiratory virus that attacks mainly the upper and lower airways, viraemia is detected in up to 40% of the patients with COVID-19, leading to extrapulmonary manifestations affecting multiple organs, including the placentas [10]. Placental infection by SARS-CoV-2, although rare, has been demonstrated in placental tissues by electron microscopy, immunohistochemistry staining (for viral nucleocapsid and spike protein), or molecular (reverse transcription-polymerase chain reaction (RT-PCR) testing, ribonucleic acid (RNA) in situ hybridisation) methods [11,12,13]. Interestingly, not all neonates with a confirmed placental SARS-CoV-2 positivity tested positive [14]. Maternal infection does not necessarily warrant placental infection. Likewise, placental viral infection does not guarantee transplacental transmission to the developing fetus [15]. The protective role of the placenta in preventing transplacental transmission of intrauterine SARS-CoV-2 infection is intriguing. The possible mechanisms include the maternal-placental-fetal interface serving as a natural immuno-mechanical barrier against pathogens or the absence of specific pathways/receptors that permit successful viral transmission [12].

Noteworthy, SARS-CoV-2 gains entry to host cells via binding to the angiotensin-converting enzyme 2 (ACE-2) receptor [16]. ACE-2 is a key regulator of the renin-angiotensin-aldosterone system (RAAS) and catalyses the conversion of angiotensin II (vasopressin) to angiotensin-(1–7) [17]. The binding of viral spike protein to endogenous ACE2 results in internalisation and downregulation of ACE2 surface expression [18]. The disruption of the critically balanced ACE/ACE2 and RAAS activation secondary to unopposed angiotensin II activity ultimately triggers the so-called “cytokine storm”, a potentially life-threatening event related to COVID-19 infection. Conversely, the abundance of ACE-2 on the placental cell surface, in particular syncytiotrophoblasts, may paradoxically limit viral infection [19].

The placenta is a diary of the pregnancy, and histopathological examination of the placenta can reveal valuable information with regard to the health of both the mother and fetus [20]. A myriad of placental histomorphological alterations had been implicated in maternal COVID-19 infection. For instance, maternal vascular malperfusion (MVM) in the form of decidual vasculopathy, accelerated villous maturation and distal villous hypoplasia with Tenney–Parker changes were exemplified in many published data [21,22]. Inflammatory condition chronic histiocytic intervillositis with associated villous trophoblast necrosis had been consistently associated with transplacental SARS-CoV-2 transmission observed in a series of stillborn infants of COVID-19-infected mothers [23]. Other histological aberrations such as fetal vascular malperfusion (FVM) and inflammatory lesions, including chorioamnionitis, funisitis and villitis) was also previously described [24,25].

To date, there is no pathognomonic morphological pattern seen in the placenta that is characteristic of COVID-19 maternal infection [26,27,28,29]. Noteworthy, the published data on placentas are limited to SARS-CoV-2-infected pregnant women thus far, lacking carefully matched controls. Whether transplacental transmission of SARS-CoV-2 occurs is still a subject of much debate and concern. The association between abnormal placental histomorphological features, disease severity and perinatal outcomes is still largely uncertain. Little is known about the time course of SARS-CoV-2 maternal infection in relation to the histomorphological changes in the placenta.

In the present study, we retrospectively analysed the histomorphological features in placentas of pregnant women infected with SARS-CoV-2, with a comparison of placentas of pregnant women (from pre-COVID-19 era) matched in a 1:1 fashion for maternal age, comorbidities and gestational age at delivery. In addition, whether these changes are correlated with the duration of SARS-CoV-2 infection to delivery, severity of maternal infection, as well as perinatal outcomes, were also determined.

## 2. Materials and Methods

### 2.1. Study Subjects Identification

This was a retrospective cohort study conducted for a period of eight months, from 1 March 2021 to 31 October 2021. The subjects were recruited at the Department of Obstetrics and Gynaecology, Universiti Kebangsaan Malaysia, Malaysia. The study and research protocol were approved by our local institutional human research ethics committee (JEP-2020-308). The study subjects included all pregnant women who delivered in our institution with a confirmed SARS-CoV-2 infection at any trimester during their pregnancy. A positive SARS-CoV-2 infection was based on the detection of SARS-CoV-2 RNA from a nasopharyngeal swab by RT-PCR test. Cases with multiple pregnancies, known fetal anomalies or whose placentas were not sent for histopathological evaluation were excluded from the study.

SARS-CoV-2 negative controls consisted of those who delivered prior to the COVID-19 pandemic. They were retrospectively selected from our hospital’s Integrated Laboratory Management System (ILMS) database. All placentas submitted for histopathological evaluation between January 2018 and December 2018 were screened in sequential order from our database in a chronological order, who matched in a 1:1 fashion with our study subject by maternal comorbidities, maternal age and gestational age at delivery, were recruited as the control group.

Clinicopathological data such as the subjects’ age, gestational age, parity, route of delivery, COVID-19 infection-to-delivery interval, clinical symptoms and disease severity were recorded. The severity of COVID-19 infection was classified into five categories based on clinical status (asymptomatic, mild, moderate, severe and critical), as previously defined based on the National Institute of Health [30] (Appendix A). Birth outcomes of the neonates (birth weight, gestational age at delivery, Apgar score at 1 and 5 min and COVID-19 test results) were also collected. Primary adverse outcomes of interest included maternal and perinatal death, small for gestational age, prematurity (birth prior to 37 weeks of gestation), and 1 and 5 min Apgar scores of 3 or less. The data obtained were anonymised as each subject was coded accordingly.

### 2.2. Placental Histopathological Examination

Placentas of these COVID-positive mothers were collected, weighed and immersed immediately into 10% buffered formalin following delivery. These placental tissues were handled in accordance with the standard operating procedures for COVID-positive specimens. All placentas were sampled according to the standard protocol as previously described [31]. Briefly, one section each from the umbilical cord, membrane, a full thickness of placenta, shaved maternal and fetal surface of the placenta and any abnormal areas were sampled. Histological sections at 4 µm, stained with haematoxylin and eosin, were evaluated by two perinatal pathologists (G.C.T. and Y.P.W.) who were blinded to clinical information of the cases. The placentas were examined following the 2016 Amsterdam Placental Workshop Group Consensus guidelines [32].

In addition, the percentage of syncytial nuclear aggregates (SNAs), inclusive of syncytial knots and bridges, against chorionic villi was assessed from the midparenchymal region of the placenta and was calculated as previously described [31]. The periphery of the basal plate regions of the placenta was excluded from evaluation. Syncytial knot was recognised as a multilayered aggregation of at least five syncytiotrophoblast nuclei that demonstrated heavily condensed chromatin and bulged slightly from the villous surface. Syncytial bridges were identified as highly nucleated regions that connect two neighbouring villi. Values from the two independent observers were averaged, giving a single value for each case.

### 2.3. Immunohistochemistry (IHC) Analysis

Formalin-fixed paraffin-embedded tissue blocks were sectioned at approximately 3 µm and mounted onto positively charged glass slides. The slides were left to be air-dried at room temperature overnight before placing on a hot plate at 60 °C for 1 h. The slides were deparaffinised with xylene and rehydrated in alcohol. IHC antigen retrieval was performed using Dako Target Retrieval Solution, High pH (code no. S1699, Dako, Glostrup, Denmark) for 30 min at 110 °C. The slides were subsequently treated with EnVision^TM^ FLEX Peroxidase-Blocking Reagent (code no. DM821, Dako, Glostrup, Denmark) for 10 min before being incubated with primary mouse monoclonal antibody SARS-CoV/SARS-CoV-2 [clone 1A9, cat no. GTX632604] at dilution 1:700 for another 30 min at room temperature. Sections were then incubated with EnVision^TM^ FLEX horseradish peroxidase-conjugated secondary antibody (code no. K8023, Dako, Glostrup, Denmark) for 30 min at 37 °C. 3,3′-diaminobenzidine (DAB) substrate was added for 10 min to visualise the immunoreactivity. The slides were then counterstained with Hematoxylin 2 (Ref. 7231, ThermoScientific, Waltham, MA, USA) and mounted using CoverSeal^TM^-X xylene-based mounting medium (cat. no. FX2176, Cancer Diagnostics, Durham, NC, USA). A known SARS-CoV-2-infected lung tissue served as the positive control.

The immunoexpression of SARS-CoV-2 on placental tissue sections was evaluated by two perinatal pathologists (G.C.T. and Y.P.W.) independently using Olympus light microscope BX40 (Olympus Corporation, Tokyo Japan), who were blinded to the sample COVID-19 status. Cytoplasmic and/or nuclear immunoexpressions for SARS-CoV-2 of any staining intensity in more than 1% of cells of interest were considered positive immunostaining. The extent and localisation of the immunostains at different regions of the placenta (syncytiotrophoblasts, cytotrophoblasts, villous endothelial cells, maternal decidua, maternal endothelial cells) were also documented.

### 2.4. Statistical Analysis

Data collected was analysed statistically using Statistical Package for the Social Sciences (SPSS) statistical software version 26.0 (PASW Statistics, Chicago, IL, USA). Categorical variables were expressed as numbers and percentages, while continuous data were expressed as mean ± standard deviation (SD). Chi-square, Fisher exact test and student *t*-test were performed to compare the differences between variables. Univariate and multivariable logistic regression models were conducted, and odds ratios were used to associate maternal SARS-CoV-2 status with undesirable maternal and neonatal outcomes while controlling for other covariates. A *p*-value of less than 0.05 was considered statistically significant.

## 3. Results

### 3.1. Maternal Characteristics, Obstetric and Neonatal Outcomes

Our study subjects consisted of 47 pregnant women who had SARS-CoV-2 infection confirmed in the second (n = 12, 25.5%) and third trimester (n = 35, 74.5%) during their pregnancy and delivered in our institution. None of them acquired COVID-19 infection in the first trimester of pregnancy. A total of 17 (36.2%) women had recovered from COVID-19 infection, while 30 (63.8%) were still in active infection at delivery. The majority (n = 33/47, 70.2%) of the study subjects were asymptomatic or had only mild symptoms, while 14 (29.8%) gravid women had moderate, severe or critical illness. Of these, four (4/14, 28.6%) succumbed due to COVID-19-related complications. The mean infection-to-delivery interval was 34.98 ± 55.08 (range, 0–215) days.

The mean maternal age at delivery was 31.45 ± 4.58 (range, 25–43) years and the mean gestational age at delivery was 36.18 ± 3.72 (range, 26.0–40.4) weeks. The majority of our study subjects were of Malay ethnicity (n = 42, 89.4%), followed by Chinese (n = 2, 4.3%) and others including Indonesian and Burmese (n = 3, 6.4%). A total of 26 (55.3%) patients had at least one comorbidity, such as hypertension (6/47, 12.8%), diabetes mellitus (14/47, 29.8%), obesity (10/47, 21.3%), anaemia in pregnancy (6/47, 12.8%), cardiac abnormalities (2/47, 4.3%) and bronchial asthma (3/47, 6.4%). Of the 47 patients, 16 (34.0%) pregnancies resulted in preterm delivery following an emergency caesarean section. The indications for emergency caesarean section were worsened maternal respiratory condition or fetal distress. General characteristics between the case and control groups were similar, thus validating that those matching variables were sufficiently paired (*p* > 0.05), except for a higher rate of caesarean section in the COVID-19 cases (*p* < 0.001). Maternal comorbidities profile, gestational age at delivery and maternal age were matched between the COVID-19 infected and the control groups to ensure appropriate comparison without confounding/bias.

The mean birth weight was 2500.26 ± 636.83 g (range, 900–3770 g); mean fetal-placental weight ratio was 4.90 ± 0.87 g (range 2.75–6.56 g); mean Apgar scores at 1 and 5 min were 7.74 ± 2.52 and 8.83 ± 2.32, respectively, which was significantly lower than that of controls (*p* = 0.035 and *p* = 0.002, respectively). Of the 47 neonates born to women with COVID-19 diagnosis, only two (4.3%) tested positive in the first 24 h and in the first 72 h after birth, respectively. Three (6.4%) neonatal deaths were recorded in this study, born prematurely following preterm delivery via emergency caesarean section owing to worsening maternal respiratory illnesses. All of these deaths coincided with unvaccinated pregnancies. These neonates nonetheless tested negative for SARS-CoV-2. Almost half (n = 20/47, 42.6%) of the neonates born to mothers with COVID-19 infection suffered from fetal growth restriction and were born small for gestational age, which was significantly higher compared with the control group (*p* = 0.025). Relevant clinicopathological characteristics of both study subjects and controls are summarised in Table 1.

We then compared the clinicopathological characteristics and maternal and perinatal outcomes between the active and resolved COVID-19 cases (Table 2). The active SARS-CoV-2 infected cases had a significantly lower gestational age at delivery (35.37 ± 4.11 weeks) and a higher caesarean section rate (*p* = 0.007) compared to the resolved cases, owing to worsening maternal respiratory condition. In addition, there were a significant lower fetal birth weight (*p* = 0.015), placenta weight (*p* = 0.032), fetal-placental weight ratio (*p* = 0.022) and poorer Apgar score (*p* < 0.001). The mean SARS-CoV-2 infection to delivery interval was 4.67 ± 5.57 days. Likewise, we observed a similar trend when comparing symptomatic with asymptomatic COVID-19 groups, with the symptomatic group exhibiting a significantly higher frequency of preterm delivery (*p* < 0.001), lower fetal-placental weight ratio (*p* = 0.040) and a worse Apgar score (*p* < 0.001). There were three (n = 3/47, 6.4%) neonatal deaths in the SARS-CoV-2 infected group, mainly related to severe prematurity (Table 2).

### 3.2. Placental Histomorphological Alteration and Immunohistochemical Findings

The results of the histopathological examination of placenta from the COVID-19 pregnant women are presented in Figure 1.

Interestingly, the percentage of syncytial knots per villous in the COVID-19-infected placentas was significantly higher (34.16 ± 12.55%) compared to the control group (31.52% ± 9.72%) (*p* = 0.022). Of note, the occurrence of decidual vasculopathy (n = 22/47; *p* = 0.004), maternal vascular thrombosis (n = 9/47; *p* = 0.015), low grade chronic villitis (n = 5/47; *p* = 0.03), chronic histiocytic intervillositis (n = 10/47; *p* = 0.027) and chorangiosis (n = 7/47; *p* = 0.012) were significantly higher in the COVID-19-infected placentas compared to the matched control group (Figure 2). Histomorphological alterations identified in both groups are summarised in Table 3.

Next, we focused further on the histomorphological alterations in placentas from women with active SARS-CoV-2 infection to those who had resolved from the illness. Other than the significant occurrence of maternal vascular thrombosis in the active group (*p* = 0.042), there were no other relevant differences in the histological characteristics between these two groups (Table 3). Similarly, we observed no significant differences in the frequencies of various placental histomorphological patterns in women with symptomatic SARS-CoV-2 infection to those who were asymptomatic (Appendix A).

Almost all of the COVID-19-positive cases were negative for SARS-CoV-2 immunohistochemistry except for one case (n = 1/47; 2.13%), which showed unequivocal cytoplasmic positivity in the syncytiotrophoblasts of the chorionic villi of the infected placenta. This placenta was from a 38-year-old mother with gestational diabetes mellitus, presented at 35 weeks of gestation with moderate symptoms of COVID-19 pneumonia, diagnosed six days prior to delivery. The placenta demonstrated chronic histiocytic intervillositis and focal intervillous thrombosis histologically. Intriguingly, the newborn tested negative for SARS-CoV-2 at birth.

### 3.3. Risk Factors for Adverse Maternal and Perinatal Outcomes in COVID-19-Infected Pregnancies

Compared with resolved COVID-19 cases, active COVID-19 cases were strongly associated with preterm birth (adjusted odds ratio (aOR) 9.23, 95% CI 1.294 to 65.83, *p* = 0.027). Similarly, symptomatic and severe-critical COVID-19 in pregnancy were associated with an elevated risk of preterm birth, compared to the milder form of COVID-19 infection (aOR 4.958, 95% CI 1.08 to 22.753, *p* = 0.039 and aOR 17.034, 95% CI 2.886 to 100.524, *p* = 0.002, respectively) in adjusted analysis. Notably, the risk for preterm delivery was increased with the presence of maternal inflammatory response in the placenta (aOR 7.113, 95% CI 1.579 to 32.033, *p* = 0.011) (Figure 3). Maternal age, comorbid, mode of delivery and other placental histomorphological changes had no significant association (*p* > 0.05) (Appendix A).

Severe-critical maternal COVID-19 had a significantly higher risk of maternal mortality than the patients with a milder form of COVID-19 infection (aOR 18.763, 95% CI 1.498 to 234.988, *p* = 0.023). Our study also revealed that placentas showing altered villous maturation were significantly associated with a poorer Apgar score of 3 or less compared to those without (aOR 21.413, 95% CI 1.677 to 273.487, *p* = 0.018) (Figure 3).

There were no significant associations between maternal COVID-19 status, maternal age, comorbid, parity, mode of delivery or placental histomorphological features with other neonatal outcomes such as small for gestational age and perinatal death (*p* > 0.05) (Appendix A).

## 4. Discussion

Adverse birth outcomes, including preterm delivery, were well documented in SARS-CoV-2 affected pregnancies. Our results were in concordance with a surveillance study conducted by MyClymont et al. [33] involving 6012 pregnancies, who revealed that active SARS-CoV-2 infection was significantly associated with an increased risk of preterm birth, even among cases with milder forms of infection. We observed a significantly higher rate of caesarean section delivery in women with symptomatic COVID-19 infection. Our findings are in agreement with Khalil et al. [34]; in their systematic review and meta-analysis, they revealed that the most common indication for caesarean section was not fetal distress, but rather related to the fear of sudden maternal deterioration following COVID-19 infection or severe maternal COVID-19 pneumonia.

In this study, we recorded 4/47 (8.5%) cases of COVID-19-related maternal death involving unvaccinated pregnant women, with all of them presented in the severe-critical stages of the disease. Our results were in agreement with Villar et al. [35], who revealed that severe pregnancy complication rates were the highest, especially if the women presented with fever and acute onset shortness of breath, reflecting systemic disease.

Our data summarised a spectrum of pathological findings observed in placentas delivered by 47 pregnant women infected with SARS-CoV-2. We found that maternal vascular malperfusion (MVM) is the most common histopathological feature observed. MVM, which encompasses accelerated villous maturation with increased syncytial knots, distal villous hypoplasia, decidual vasculopathy, maternal vascular thrombosis and villous infarction, is frequently described in placentas of pregnant women with hypertensive disorders [32]. Of the 36 cases demonstrating MVM, interestingly, only six patients were found to be hypertensive. We observed a significantly higher rate of decidual vasculopathy in the placentas of women with COVID-19 infection compared to the age and comorbidities matched controls, in consonance with the previously reported study [27].

Besides decidual vasculopathy, maternal vascular thrombosis was significantly more frequent in the active COVID-19 group. Accumulating evidence suggests that COVID-19 is associated with an aberrant or exaggerated host inflammatory response with a sudden surge of pro-inflammatory cytokines, recognised as ‘cytokine storm’. Such inflammatory alterations contribute to vascular endothelial dysfunction and subsequent derangements in the coagulation system, resulting in hypercoagulability with elevated D-dimer levels and microthrombi formation [36].

Additionally, we observed a significant increase in the syncytial knots in the COVID-19 infected group—this feature is widely referred to as Tenney–Parker changes. Syncytial knots reflect villous maturity when increased and are regarded as a placental response to trophoblastic ischaemia/underperfusion or maternal hypoxia, classically seen in pre-eclampsia [37,38,39] and some maternal bacterial infections [28,31,40,41]. Shchegolev et al. [42] revealed that the number of syncytial knots in placental villi depended on COVID-19 disease severity, with a corresponding increase in VEGF immunoexpression in the syncytiotrophoblasts and villous endotheliocytes. Comparably, we detected a higher rate of syncytial knot formation in the symptomatic and active COVID-19 groups than in the asymptomatic and resolved counterparts; however, it was not statistically significant.

Likewise, we observed that placentas showing accelerated villous maturation characterised by hypermature, hypoplastic slender terminal villi and increased syncytial knots posted a significant risk for poor Apgar scores of 3 or less at birth compared to those without. Retardation in villous tree maturation can cause diminution of vasculosyncytial membranes resulting in placental insufficiency and fetal hypoxia and is associated with considerable neonatal morbidity and mortality [43]. Chorangiosis, which represents an adaptive response to chronic placental underperfusion/hypoxia, is a lesion frequently described in the COVID-19 literature [44], which is similar to our study cohort.

Although many reported cases of newborns born to mothers with positive SARS-CoV-2 infection tested positive, our data reaffirms that vertical transmission via the transplacental route was rare. Of the 47 cases, only one of the placentas was demonstrated positive for SARS-CoV-2 by the immunohistochemistry method, corroborating the suggestion of potential transplacental maternal-fetal transmission. It is noteworthy that these cases also had diabetes mellitus. Our previous report found that candida chorioamnionitis in two cases of pregnancies complicated with diabetes mellitus [45]. The question of whether diabetes mellitus weakens the immune status and facilitates the infection is to be determined. In line with other studies, the virus was found localised within the cytoplasm of the syncytiotrophoblasts. Other placental cell types that could be involved with the SARS-CoV-2 virus are cytotrophoblasts, Hofbauer cells and endothelial cells [46].

Recent reports, including ours, revealed that inflammatory alterations, particularly chronic histiocytic intervillositis with trophoblast necrosis (also known as SARS-CoV-2 placentitis), represented a pattern of injury associated with transplacental SARS-CoV-2 transmission [46]. It remains unclear if the destruction of syncytiotrophoblasts is a sequela of the direct viral cytopathic effect or is secondary to inflammatory injury. Nonetheless, damage to this protective barrier allows viral access to the villous core, which facilitates fetal infection.

Viruses associated with TORCH pathogens (*Treponema pallidum*, *Toxoplasma gondii*, rubella virus, cytomegalovirus) are notorious for the cause of chronic villitis in placentas [47]. Similar to our findings, Patberg et al. [48] observed a three times higher incidence of chronic villitis/villitis of unknown aetiology (VUE) in the COVID-19 cases than expected in normal term placentas. The exact mechanisms behind VUE remain uncertain; excessive systemic immune responses following cytokine storms may cause direct or indirect immune damage to the placentas [49].

Expectedly, our study demonstrated that maternal inflammatory response was significantly associated with seven times increased risk of preterm delivery among the COVID-19-infected group, in line with other studies [50,51]. Inflammation has long been implicated in the pathogenesis of preterm and term labour [52]. Elevated circulating IL-6 level was linked with severe SARS-CoV-2 infection and may play a pivotal role in the activation of a series of mediators and inflammatory pathways leading to preterm delivery [51].

Other pathological findings, including fetal vascular malperfusion, villous infarction and maternal and fetal inflammatory responses, were also observed in the present study. However, they did not reach statistical significance when comparing the COVID-19 infected to the control group.

The newborn outcomes were directly linked to maternal health status. We reported three cases of neonatal death following severe maternal COVID-19 disease in our study cohort. Thus, not surprisingly, pregnant women with active and symptomatic SARS-CoV-2 infections are more likely to exhibit undesirable fetal outcomes. The majority of the asymptomatic or resolved COVID-19 pregnant women had term deliveries without reported complications. Moreover, our findings also suggested that SARS-CoV-2 infection during pregnancy can result in placental damage, thereby triggering local inflammatory processes at the maternal-fetal interface and subsequent reduced placental blood flow. Functional impairment of the placenta caused by SARS-CoV-2 infection, as was recently described [53], increases the risk of fetal compromise, as evidenced by a higher frequency of fetal demise, lower birth weight, and poorer Apgar score recorded in the active COVID-19 cases.

We identified several limitations in this study, most of which were inherent in conducting retrospective clinicopathological research. Existing clinicopathological data and patients’ comorbid and documented neonatal outcomes may be incomplete. For instance, we were unable to extract complete data with regards to maternal body mass index and cord pH—an important indicator for the wellbeing of a newborn. Our study was also limited by the number of maternal COVID-19 cases presented to our centre for delivery. A large-scale multi-centre study with a larger number of study subjects would be able to provide a clearer picture of the placental histomorphological features associated with SARS-CoV-2 transplacental infection.

## 5. Conclusions

In summary, adverse maternal and neonatal outcomes are more likely to occur in pregnant women with active, symptomatic and severe-critical SARS-CoV-2 infections. A myriad of pathological conditions has been documented in the placentas induced by maternal COVID-19 infection, with maternal vascular malperfusion being the most common. Notably, our study revealed that the placentas of COVID-19-infected mothers had increased SNAs and were at a significantly higher risk of developing maternal vascular thrombosis. Altered villous maturation and severe-critical maternal COVID-19 infection were associated with an elevated risk of poor Apgar scores at birth and maternal mortality, respectively.

## Figures and Tables

**Figure 1 ijerph-19-09517-f001:**
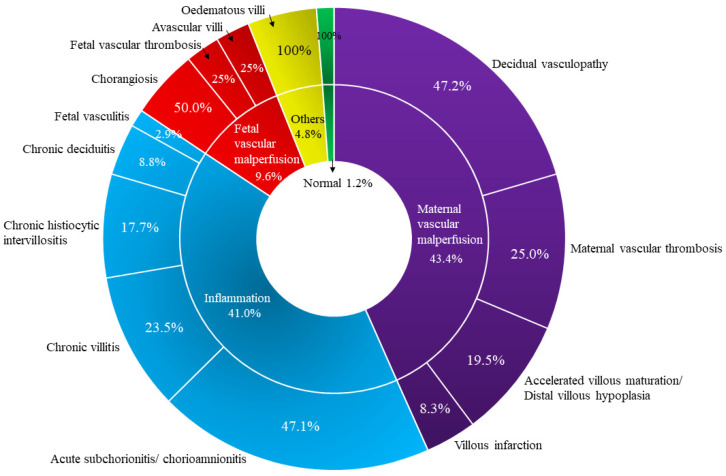
Histological characteristics of SARS-CoV-2-infected placentas (n = 47).

**Figure 2 ijerph-19-09517-f002:**
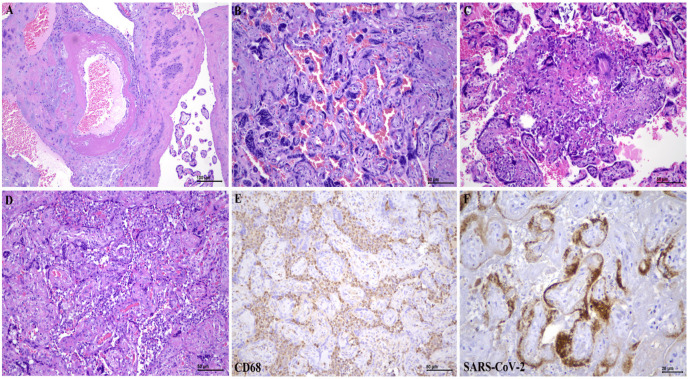
Histopathological features of placentas in SARS-CoV-2 infected patients. (**A**) Maternal arteriole with atherosis (H&E ×100). (**B**) Tenney Parker changes: increase in syncytial knotting in a 32-week placenta (H&E ×200). (**C**) Chronic villitis/villitis of unknown aetiology (H&E ×200). (**D**) Chronic histiocytic intervillositis (H&E ×200) with (**E**) immunohistochemistry with CD68 highlighting the presence of histiocytes within the intervillous space (CD68, ×200) and (**F**) immunohistochemistry with SARS-CoV-2 spike protein demonstrating the presence of viral protein within cytoplasms of syncytiotrophoblasts (SARS-CoV-2, ×400).

**Figure 3 ijerph-19-09517-f003:**
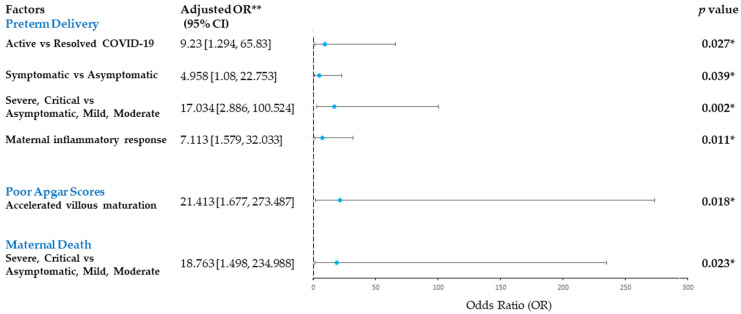
Forest plot showing adjusted logistic regression model to assess the association between maternal COVID-19 status and histopathological features with adverse maternal and neonatal outcomes, i.e., maternal death, poor Apgar scores and preterm delivery. Odds ratio (OR) and 95% confidence intervals (CI) are depicted. * statistically significant; ** adjusted for maternal age and comorbid.

**Table 1 ijerph-19-09517-t001:** Clinicopathological characteristics, maternal and perinatal outcomes of COVID-19-infected pregnant women and the control groups.

Clinicopathological Features		COVID-19 Cases n = 47 (%)	Controls n = 47 (%)	*p*-Value
Maternal age (years)		31.45 ± 4.58	31.47 ± 3.68	0.158
Gestational age (weeks)		36.18 ± 3.72	36.38 ± 3.69	0.931
Ethnicity	Malay	42 (89.3)	46 (97.9)	0.235
	Chinese	2 (4.3)	0 (0.0)	
	Others	3 (6.4)	1 (2.1)	
Delivery mode	Caesarean section	37 (78.7)	17 (36.2)	<0.001 *
	Assisted delivery	2 (4.3)	0 (0.0)	
	Vaginal delivery	8 (17.0)	30 (63.8)	
Comorbidity	No	22 (46.8)	21 (44.7)	1.000
	1 comorbid	17 (36.2)	14 (29.8)	
	More than 1 comorbid	8 (17.0)	12 (25.5)	
Severity of COVID-19	Asymptomatic	19 (40.4)	N/A	N/A
	Mild	14 (29.8)		
	Moderate	5 (10.6)		
	Severe	7 (14.9)		
	Critical	2 (4.3)		
Maternal death	Yes	4 (8.5)	0 (0.0)	0.117
	No	43 (91.5)	47 (100.0)	
Infection-to-delivery interval (days)		34.98 ± 55.08	N/A	N/A
Newborn birth weight (grams)		2500.26 ± 636.83	2557.32 ± 703.10	0.339
Fetal growth restriction/small for gestation		20 (42.6)	9 (19.1)	0.025 *
Placental weight (grams)		509.68 ± 108.45	507.67 ± 113.72	0.595
Fetal-placental weight ratio		4.90 ± 0.87	5.13 ± 1.21	0.083
Apgar score (1 min)		7.74 ± 2.52	7.94 ± 1.74	0.035 *
Apgar score (5 min)		8.83 ± 2.32	9.23 ± 1.03	0.002 *
Neonatal death	Yes	3 (6.4)	1 (2.1)	0.617
	No	44 (93.6)	46 (97.9)	

* statistically significant between COVID-19-infected mothers and the control group; N/A, not applicable.

**Table 2 ijerph-19-09517-t002:** Clinicopathological features and outcomes of COVID-19 active vs. resolved cases as well as asymptomatic vs. symptomatic cases.

Clinicopathological Features		COVID-19 Cases			COVID-19 Cases		
		Active n = 33 (%)	Resolved n = 14 (%)	*p*-Value	Asymptomatic n = 19 (%)	Symptomatic n = 28 (%)	*p*-Value
Maternal age (years)		32.18 ± 4.10	29.71 ± 5.31	0.274	32.42 ± 4.97	30.79 ± 4.26	0.776
Gestational age (weeks)		35.37 ± 4.11	38.08 ± 1.38	<0.001 *	37.68 ± 1.24	35.16 ± 4.46	<0.001 *
Ethnicity	Malay	29 (87.9)	13 (92.9)	0.427	18 (94.7)	24 (85.7)	0.376
	Chinese	1 (3.0)	1 (7.1)		1 (5.3)	1 (3.6)	
	Others	3 (9.1)	0 (0.0)		0 (0.0)	3 (10.7)	
Delivery mode	Caesarean section	29 (87.9)	8 (57.1)	0.007 *	15 (78.9)	22 (78.6)	1.000
	Assisted delivery	2 (6.1)	0 (0.0)		1 (5.3)	1 (3.6)	
	Vaginal delivery	2 (6.1)	6 (42.9)		3 (15.8)	5 (17.9)	
Comorbidity	No	16 (48.5)	5 (35.7)	0.470	7 (36.8)	14 (50.0)	0.601
	1 comorbid	8 (24.2)	6 (42.9)		7 (36.8)	7 (25.0)	
	More than 1 comorbid	9 (27.3)	3 (21.4)		5 (26.3)	7 (25.0)	
Severity of COVID-19	Asymptomatic	14 (42.4)	5 (35.7)	0.124	N/A	N/A	N/A
	Mild	8 (24.2)	6 (42.9)				
	Moderate	2 (6.1)	3 (21.4)				
	Severe	7 (21.2)	0 (0.0)				
	Critical	2 (6.1)	0 (0.0)				
Maternal death	Yes	4 (12.1)	0 (0.0)	0.302	0 (0.0)	4 (14.3)	0.137
	No	29 (87.9)	14 (100.0)		19 (100.0)	24 (85.7)	
Infection-to-delivery interval (days)		4.67 ± 5.57	106.43 ± 53.18	<0.001 *	39.53 ± 68.60	31.89 ± 44.80	0.054
Newborn birth weight (g)		2443.09 ± 728.11	2635.00 ± 319.71	0.015 *	2708.95 ± 416.29	2358.64 ± 723.85	0.054
Placental weight (g)		504.09 ± 125.23	522.86 ± 52.39	0.032 *	518.68 ± 64.72	503.57 ± 130.96	0.012 *
Fetal-placental weight ratio		4.84 ± 0.97	5.06 ± 0.59	0.022 *	5.23 ± 0.60	4.68 ± 0.96	0.040 *
Apgar score (1 min)		7.24 ± 2.87	8.93 ± 0.27	<0.001 *	8.63 ± 1.17	7.14 ± 3.00	<0.001 *
Apgar score (5 min)		8.39 ± 2.65	9.86 ± 0.36	<0.001 *	9.68 ± 0.75	8.25 ± 2.81	<0.001 *
Neonatal death	Yes	3 (9.1)	0 (0.0)	0.544	0 (0.0)	3 (10.7)	0.262
	No	30 (90.9)	14 (100.0)	0.274	19 (100.0)	25 (89.3)	

* statistically significant between active COVID-19 and resolved cases as well as between asymptomatic and symptomatic cases; N/A, not applicable.

**Table 3 ijerph-19-09517-t003:** Histological features of placentas from COVID-19-infected cases vs. controls as well as active vs. resolved COVID-19 cases.

Histological Features		COVID-19 Cases n = 47 (%)	Controls n = 47 (%)	*p*-Value	COVID-19 Cases		*p*-Value
					Active n = 33 (%)	Resolved n = 14 (%)	
Maternal vascular malperfusion	Accelerated villous maturation/distal villous hypoplasia	8 (17.0)	7 (14.9)	1.000	7 (21.2)	0 (0.0)	0.086
	Syncytial knots (mean ± SD)	34.16 ± 12.55	31.52 ± 9.72	0.022 *	35.17 ± 13.62	31.77 ± 9.61	0.091
	Villous infarction	3 (6.4)	6 (12.8)	0.486	3 (9.1)	3 (21.4)	0.344
	Decidual vasculopathy	22 (46.8)	8 (17.0)	0.004 *	17 (51.5)	5 (35.7)	0.358
	Maternal vascular thrombosis	9 (19.1)	1 (2.1)	0.015 *	9 (27.3)	0 (0.0)	0.042 *
Fetal vascular malperfusion	Avascular villi	3 (6.4)	0 (0.0)	0.242	2 (6.1)	1 (7.1)	1.000
	Fetal vascular thrombosis	3 (6.4)	0 (0.0)	0.242	2 (6.1)	1 (7.1)	1.000
	Chorangiosis	7 (14.9)	0 (0.0)	0.012 *	4 (12.1)	3 (21.4)	0.410
Chronic inflammation	Chronic deciduitis	5 (10.6)	1 (2.1)	0.203	3 (9.1)	2 (14.3)	0.627
	Chronic villitis, low grade	5 (10.6)	1 (2.1)	0.030 *	5 (15.2)	0 (0.0)	0.193
	Chronic villitis, high grade	6 (12.8)	1 (2.1)		3 (9.1)	3 (21.4)	
	Chronic histiocytic intervillositis	10 (21.3)	2 (4.3)	0.027 *	6 (18.2)	4 (28.6)	0.456
Maternal inflammatory response	Acute subchorionitis	21 (44.7)	11 (23.4)	0.056	15 (45.5)	6 (42.9)	0.765
	Acute chorioamnionitis	2 (4.3)	6 (12.8)		1 (3.0)	1 (7.1)	
Fetal inflammatory response	Chorionic vasculitis &/umbilical phlebitis	4 (8.5)	3 (6.4)	0.255	1 (3.0)	2 (14.3)	0.208
	Umbilical arteritis	3 (6.4)	0 (0.0)		0 (0.0)	0 (0.0)	
Others	Oedematous villi	5 (10.6)	9 (19.1)	0.386	4 (12.1)	1 (7.1)	1.000

* statistically significant between COVID-19-infected mothers and the control group as well as between active COVID-19 and resolved cases.

## Data Availability

Not applicable.

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
