# Peer review of "SARS-CoV-2 Infection in Pregnancy: Placental Histomorphological Patterns, Disease Severity and Perinatal Outcomes"

_ijerph, 2022, doi:10.3390/ijerph19159517_

Round 1

Reviewer 1 Report

Dear authors:   First of all, I would like to congratulate you on your research work. It is an interesting and interesting and topical, and I would like to make some suggestions for improvement.   - The section on material and methods should be improved, as it is not very clear which groups are involved. involved. Also, it is not clear whether patients have completed informed consent to participate in this study. to participate in this study. - Also, I suggest that in the discussion section you reinforce the current literature and include a section on the limitations of your study. section on the limitations of your study. - Tables 1 and 2 are cut across several pages, please edit this format so that they can be spread over one page. spread over one page. - I would like to congratulate you on the images of slits 1 and 2.   Best regards.

Author Response

Dear authors:   First of all, I would like to congratulate you on your research work. It is an interesting and interesting and topical, and I would like to make some suggestions for improvement.   - The section on material and methods should be improved, as it is not very clear which groups are involved. 

Our responses:  Thank you for the comments. The study involved two groups of study subjects: (1) all pregnant women who delivered in our institution with a confirmed SARS-CoV-2 infection (who fulfilled all inclusion and exclusion criteria); and (2) SARS-CoV-2 negative controls, who delivered prior to the COVID-19 pandemic (see lines 128 – 135).

Inclusion/exclusion criteria for participants in both the COVID-19 and control groups were also included for further clarification (see lines 128 - 135).

Also, it is not clear whether patients have completed informed consent to participate in this study. to participate in this study. –

Our responses:  Thank you for the comment. This is a retrospective study looking at the clinicopathological data/records from all pregnant women with COVID-19 that have had presented to our institution. We used archival placental tissue samples and all patients’ information were anonymised. Hence, informed consent was not required for such study. We have had acquired ethical approval from our institutional human research ethics committee board (JEP-2020-308) prior commencing the study.

Also, I suggest that in the discussion section you reinforce the current literature and include a section on the limitations of your study. 

Our responses:  Thank you for the suggestion. We had added in a section describing the limitations of our study in the discussion. See lines 456 – 464.

- Tables 1 and 2 are cut across several pages, please edit this format so that they can be spread over one page.  - I would like to congratulate you on the images of slits 1 and 2.   Best regards.

Our responses:  Thank you for the comment. We had edited the Tables accordingly. See Table 1, line 255 and Table 2, line 272.

Reviewer 2 Report

Dear authors,

This is a very interesting manuscript. I have some suggestions and comments.

Introduction - Very well written

Materials and Methods - Line 141 - please define ILMS

Results - Line 213-214 - It is 4/11 or 4/12 pregnant women who died? Were these mothers who died from the comorbidity group?

-Line 212-213 - these data did not match with the ones in Table 1. Please check. 

Discussion: - Line 323 ".No maternal death was however observed in the control group." - This may be a selection bias. I suggest removing this phrase. 

Author Response

Dear authors,

This is a very interesting manuscript. I have some suggestions and comments.

Introduction - Very well written

Materials and Methods - Line 141 - please define ILMS

Our responses:  Thank you for the comment. ILMS was spelled in its full form. See line 135.

Results - Line 213-214 - It is 4/11 or 4/12 pregnant women who died? Were these mothers who died from the comorbidity group?

Our responses:  Thank you for pointing that out. Yes, it should be 4/14 pregnant women who succumbed. We had made the correction accordingly. See line 216.

-Line 212-213 - these data did not match with the ones in Table 1. Please check. 

Our responses:  Thank you for pointing that out. We had made the correction accordingly. See lines 215-216.

Discussion: - Line 323 ".No maternal death was however observed in the control group." - This may be a selection bias. I suggest removing this phrase. 

Our responses:  Thank you for the comment. We removed the phrase as suggested. See line 369.

Reviewer 3 Report

Wong P et al conducted a study to investigate clinical characteristics, birth outcomes and histomorphological features in the placentas of pregnant women infected with SARS-CoV-2 compared with those of carefully matched controls. The introduction provides very detailed scientific background context supporting this study. However, there are a several major issues that I noted in this manuscript that I strongly recommend the authors should be addressing.

1. Abstract.

-The abstract is a bit long. Please consider modifying it by including only relevant data to fit within 200 words. Also in the abstract you should not include headings.

2. Methods.

- In the methods section, in the study design, the authors state that they conducted a retrospective cohort study. However, this study appears to be a case-control study. The authors state in the introduction and in the methods that they included a control group of pregnant women who gave birth pre-pandemic.  Thus, please present more explicitly the inclusion/exclusion criteria for participants in both the COVID-19 and control groups.

- Please define the criteria for classifying COVID-19 as mild, moderate, severe and critical illness.

- The authors collected data on the histopathological characteristics of the placentas. I consider this to be the primary outcome of interest. I consider that this should be mentioned in the methods.

-Please describe how the results of the continuous variables (mean or median) were reported.

3. Results

-In the methods the authors stated that they compared the continuous variables with the t-test. However, in the results section the continuous variable data are presented as median (interquartile range). Thus for comparison purposes the t-test cannot be used. If the results are indeed reported at the median, consider comparing with a specific test for non-parametric data (e.g. Mood's median test). In addition in Table 1 continuous variables appear to be presented as mean ±SD. It is therefore necessary to describe how the continuous data were reported (mean±SD or median(IQR). The whole article shows this confusion.

- In your methods you stated that you used logistic regression to identify risk factors for neonatal adverse outcomes. However, this analysis seems to be missing from the results section. Please present the odds ratio for adverse outcomes by risk factors if logistic regression was performed.

Author Response

Wong P et al conducted a study to investigate clinical characteristics, birth outcomes and histomorphological features in the placentas of pregnant women infected with SARS-CoV-2 compared with those of carefully matched controls. The introduction provides very detailed scientific background context supporting this study. However, there are a several major issues that I noted in this manuscript that I strongly recommend the authors should be addressing.

  1. Abstract.

-The abstract is a bit long. Please consider modifying it by including only relevant data to fit within 200 words. Also in the abstract you should not include headings.

Our responses:  Thank you for the comment. We had shortened the abstract to 218 words and removed the headings as suggested.

  1. Methods.

- In the methods section, in the study design, the authors state that they conducted a retrospective cohort study. However, this study appears to be a case-control study.

Our responses:  Thank you for the comment. Although it did appear like a case control study with a carefully matched control, here we conducted a retrospective cohort study. In this study, we were looking at all COVID-19-exposed pregnancies if they brought poorer perinatal and obstetric outcomes and compared that with a carefully matched non-COVID-19-exposed pregnancies. We were not studying cases with and without poor perinatal and obstetric outcomes (from the beginning) and looked back at whether they were exposed to COVID-19/other risk factors during their pregnancies. (Please see attached simplified figure (in the attached word document) – highlighting the difference between retrospective cohort study vs case control study)

The authors state in the introduction and in the methods that they included a control group of pregnant women who gave birth pre-pandemic.  Thus, please present more explicitly the inclusion/exclusion criteria for participants in both the COVID-19 and control groups.

Our responses:  Thank you for the comment. Inclusion/exclusion criteria for participants in both the COVID-19 and control groups were included (see lines 128 - 135).

Inclusion and exclusion criteria:

COVID-19 positive group:

Inclusion criteria: All pregnant women with COVID-19 positive that were confirmed by RT-PCR, presented to our hospital between March 1, 2021 to October 31, 2021. See page 3, line 128 – 129.

Exclusion criteria: Multiple pregnancy and cases with known fetal anomalies, or whose placentas were not sent for histopathological evaluation. See page 3, lines 131 – 133.

Control group:

Inclusion criteria: Pre-COVID era, pregnant women between presented to our hospital between January 2018 and December 2018, with matched maternal age, gestational age and co-morbidities. See line 134.

- Please define the criteria for classifying COVID-19 as mild, moderate, severe and critical illness.

Our responses:  Thank you for the comment. The criteria for classifying COVID-19 were added as Supplementary Table S1. See Supplementary Table S1.

- The authors collected data on the histopathological characteristics of the placentas. I consider this to be the primary outcome of interest. I consider that this should be mentioned in the methods.

Our responses:  Yes. Placental histopathological changes were one of the focus in this study, other than maternal and perinatal outcomes in COVID-19 infected pregnant mothers. This has been described in detail in methodology. See lines 154 – 163.

-Please describe how the results of the continuous variables (mean or median) were reported.

Our responses:  Thank you for the comment. We agreed that the results of the continuous variables should be reported using mean in this study. We had made the corrections accordingly. See lines 200 - 202, 218, 220, 221 and 235.

  1. Results

-In the methods the authors stated that they compared the continuous variables with the t-test. However, in the results section the continuous variable data are presented as median (interquartile range). Thus for comparison purposes the t-test cannot be used. If the results are indeed reported at the median, consider comparing with a specific test for non-parametric data (e.g. Mood's median test). In addition in Table 1 continuous variables appear to be presented as mean ±SD. It is therefore necessary to describe how the continuous data were reported (mean±SD or median(IQR). The whole article shows this confusion.

Our responses:  Thank you for the suggestion. We agreed that the results of the continuous variables should be reported and analysed using mean, instead of median, to avoid confusion. We had made the corrections accordingly. See lines 200 - 202, 218, 220, 221 and 235.

- In your methods you stated that you used logistic regression to identify risk factors for neonatal adverse outcomes. However, this analysis seems to be missing from the results section. Please present the odds ratio for adverse outcomes by risk factors if logistic regression was performed.

Our responses:  Thank you for pointing that out. We included a section, a figure and a supplementary table to describe our analysis on various clinicopathological variables including COVID-19 severity with maternal and perinatal outcomes (using logistic regression). Discussion and references were also updated accordingly. See lines 328 – 348, lines 400 – 405, lines 433 – 439, Figure 3 and Supplementary Table S3.

Round 2

Reviewer 3 Report

The authors have provided responses to most of the questions. I am in favour of publishing.